# COVID-19 Infection Was Associated with the Functional Outcomes of Hip Fracture among Older Adults during the COVID-19 Pandemic Apex

**DOI:** 10.3390/medicina59091640

**Published:** 2023-09-11

**Authors:** Hua-Yong Tay, Wen-Tien Wu, Cheng-Huan Peng, Kuan-Lin Liu, Tzai-Chiu Yu, Ing-Ho Chen, Ting-Kuo Yao, Chia-Ming Chang, Jian-Yuan Chua, Jen-Hung Wang, Kuang-Ting Yeh

**Affiliations:** 1Department of Orthopedics, Hualien Tzu Chi Hospital, Buddhist Tzu Chi Medical Foundation, Hualien 97002, Taiwan; nonametay@gmail.com (H.-Y.T.); timwu@tzuchi.com.tw (W.-T.W.); peng0913@tzuchi.com.tw (C.-H.P.); knlnliu@tzuchi.com.tw (K.-L.L.); feyu@tzuchi.com.tw (T.-C.Y.); ihchen@tzuchi.com.tw (I.-H.C.); tkyao0318@me.com (T.-K.Y.); newfresheric@gmail.com (C.-M.C.); chua.jian.yuan@hotmail.com (J.-Y.C.); 2Department of Medical Education, Hualien Tzu Chi Hospital, Buddhist Tzu Chi Medical Foundation, Hualien 97002, Taiwan; 3School of Medicine, Tzu Chi University, Hualien 970374, Taiwan; 4Department of Medical Research, Hualien Tzu Chi Hospital, Buddhist Tzu Chi Medical Foundation, Hualien 97002, Taiwan; paulwang@tzuchi.com.tw; 5Graduate Institute of Clinical Pharmacy, Tzu Chi University, Hualien 970374, Taiwan

**Keywords:** COVID-19, hip fracture, postoperative complications, activities of daily living, mortality

## Abstract

*Background and Objectives*: Hip fractures are associated with mortality and poor functional outcomes. The COVID-19 pandemic has affected patterns of care and health outcomes among fracture patients. This study aimed to determine the influence of COVID-19 infection on hip fracture recovery. *Materials and Methods*: We prospectively collected data on patients with hip fractures who presented at Hualien Tzu Chi Hospital between 9 March 2022 and 9 September 2022. The data included demographic information and functional scores taken before, during, and after surgery. The patients were divided into two groups: COVID-19 (+) and COVID-19 (−). *Results*: This study recruited 85 patients, 12 of whom (14.12%) were COVID-19 (+). No significant differences in preoperative or perioperative parameters between the two groups were observed. The postoperative Barthel index score was significantly impacted by COVID-19 infection (*p* = 0.001). The incidence of postoperative complications was significantly correlated with general anesthesia (*p* = 0.026) and the length of stay (*p* = 0.004) in hospital. Poor postoperative functional scores were associated with lower preoperative Barthel index scores (*p* < 0.001). Male sex (*p* = 0.049), old age (*p* = 0.012), a high American Society of Anesthesiologists grade (*p* = 0.029), and a high Charlson comorbidity index score (*p* = 0.028) were associated with mortality. *Conclusions*: Hip fracture surgeries were not unduly delayed in our hospital during the COVID-19 pandemic, but the patients’ postoperative Barthel index scores were significantly influenced by COVID-19 (+). The preoperative Barthel index score may be a good predictive tool for the postoperative functional recovery of these patients.

## 1. Introduction

Hip fracture is a medical condition associated with mortality and poor functional outcomes. According to the International Osteoporosis Foundation, 1.6 million hip fractures were recorded worldwide in 2000, and this number is expected to increase to 4.5–6.3 million by 2050 because of the ageing of the population [1,2]. The mortality risk from a hip fracture persists beyond 5 years, and the 1-year mortality rate is estimated to be 20–24% [3,4]. Among patients who experience hip fractures, 40% cannot walk independently, 60% require assistance, and 33% are completely dependent or living in a nursing home 1 year after they experience the hip fracture [3,5,6].

COVID-19 is an infectious disease transmitted through air droplets and small airborne particles. The first known COVID-19 infection occurred on 31 December 2019 in China [7]. The COVID-19 pandemic was a burden on healthcare systems worldwide. Orthopedic clinics performed fewer elective and nonelective surgeries during the COVID-19 pandemic. However, orthopedic surgeries for trauma cases were still performed. The number of older adult patients presenting to hospitals with hip fractures was the same during the COVID-19 pandemic as it was before the pandemic, even in regions that experienced severe outbreaks of the virus [8]. Moldovan et al. conducted a multicenter study in Romania in 2023 and found that the COVID-19 pandemic has had a severe impact on the volume of elective arthroplasty cases in Romania’s 120 hospitals, with a dramatic decrease in the volume of primary interventions for hip and knee patients of up to 69.14% and a corresponding decline in the quality of patient care [9]. The COVID-19 pandemic has influenced patterns of care and health outcomes among patients with hip fractures [10].

In Taiwan, the COVID-19 pandemic reached a critical stage in March 2022. To prevent the spread of the virus, the government implemented strict hospital care and visitation policies. These policies influenced the quality of care, including daily living care and bedside rehabilitation, received by patients with hip fractures. An inferior quality of care has been associated with increased mortality, adverse outcomes, and poor functional improvement.

We hypothesized that the COVID-19 pandemic has led to the launch of policies that could affect hip fracture postoperative care. This includes nursing care and family care, where a decrease in the number of early bedside rehabilitation programs may further increase mortality and poor functional outcomes. Our study aimed to compare mortality and functional outcomes between patients with and without COVID-19 after surgery for hip fractures in our medical center during the COVID-19 pandemic apex in Taiwan.

## 2. Materials and Methods

### 2.1. Study Design and Population

We conducted this prospective cohort study in a single medical center in eastern Taiwan, enrolling patients aged 60 and above who were diagnosed with hip fractures between 9 March 2022 and 9 September 2022. The patients with pertrochanteric fractures received open reduction and internal fixation with a cephalomedullary nail, while the patients with displaced femoral neck fractures received hemiarthroplasty. Patients with high-energy trauma, periprosthetic or peri-implant fractures, as well as those who did not undergo surgery or underwent revision surgery for a prior hip fracture, were excluded from the study. The patients’ data were included, collected, and classified, as shown in Figure 1.

### 2.2. Data Collection

Demographic and clinical data were meticulously extracted from electronic medical records. Preoperative functional scores were assessed upon patient admission through validated tools. Follow-up postoperative functional scores were obtained during scheduled outpatient consultations or structured telephone interviews conducted by trained medical staff. The parameters were as listed as below: (1) preoperative data: age, gender, body mass index, fracture type, American Society of Anesthesiologists grade, blood chemistry, Charlson comorbidity index, Barthel index score (activities of daily living), Eastern Cooperative Oncology Group (ECOG) score, and COVID-19 status, as confirmed using polymerase chain reaction tests; (2) operative data: surgical approach, anesthesia type, use of a nerve block, preoperative NPO time, time from admission to operation, the duration of the operation, anesthesia time, estimated blood loss, and immediate postoperative hemoglobin levels; (3) postoperative data: the length of hospital stay, complication incidence (e.g., pneumonia, sepsis, acute urinary retention, ileus, electrolyte imbalance, urinary tract infection, anemia, moderate to severe hip pain), mortality, and postoperative functional scores (ECOG, Barthel index, and modified Harris hip score).

### 2.3. Statistical Analysis

All statistical analyses were performed using SPSS version 23.0 (IBM, Armonk, NY, USA). The descriptive statistics included continuous variables presented as means ± standard deviations and categorical variables shown as numbers and percentages. The univariate analysis included Student’s *t*-test, employed for comparing continuous variables between groups, and chi-square tests, used for categorical variables. The multivariable analysis included a multiple linear regression analysis conducted to assess the impact of the covariates and postoperative comorbidities on changes in the ECOG, Barthel index, and Harris hip scores, with adjustments made for potential confounders. The variable selection was based on a stepwise backward elimination process. The outcome analysis included logistic regression models used to identify factors independently associated with mortality, adjusted for confounding variables. Odds ratios and 95% confidence intervals were reported for all the logistic regression analyses. Statistical significance was defined as a *p*-value of <0.05, and all tests were two-tailed.

## 3. Results

This study comprised 85 patients, 12 of whom (14.12%) were COVID-19-positive. There were 29 (34.1%) males and 56 (65.9%) females with a mean age of 79.42 years and mean body mass index of 22.58 ± 3.72 kg/m^2^ (Table 1). In total, 39 (45.9%) of them had femoral pertrochanteric fractures and 46 (54.1%) had femoral neck fractures (Table 1). No statistically significant differences in age, body mass index, gender, ASA grade, fracture type, Charlson comorbidity index score, comorbidity status, preoperative blood chemistry data, or preoperative Barthel index score were observed between the COVID-19-positive and -negative groups (Table 1).

The COVID-19-negative group had a shorter anesthesia time than the positive group (137.73 ± 32.30 vs. 158.25 ± 40.24 min, *p* = 0.046; Table 2). No statistically significant differences in the surgical method, anesthesia method, use of perioperative nerve block, preoperative NPO time, time to operation, operation time, blood loss, or hemoglobin level were observed between the COVID-19-positive and -negative groups (Table 2). The postoperative Barthel index scores were significantly lower in the COVID-19-positive group (74.58 ± 17.64) than in the negative group (88.03 ± 16.82) (*p* = 0.014; Table 3). No significant differences in the length of hospital stay, incidence of complications, ECOG status, Harris hip score, or mortality rate were observed between the groups (Table 3).

Subgroup analyses were performed to investigate the relationships between postoperative complications and functional outcomes after 6 months in terms of the ECOG, Barthel index, and Harris hip score. A higher incidence of complications was associated with general anesthesia (β: 0.50, 95% confidence interval (CI): 0.06–0.94, *p* = 0.026) and a longer length of hospital stay (β: 0.04, 95% CI: 0.01–0.06, *p* = 0.004) (Table 4). Poor ECOG status was associated with a lower preoperative Barthel index score (β: −0.03, 95% CI: −0.04 to −0.02, *p* < 0.001) (Table 4). A lower postoperative Barthel index score was correlated with COVID-19 infection (β: −15.59, 95% CI: −24.29 to −6.89, *p* = 0.001), a higher Charlson comorbidity index score (β: −3.30, 95% CI: −5.18 to −1.42, *p* = 0.001), and a lower preoperative Barthel index score (β: 0.65, 95% CI: 0.43–0.87, *p* < 0.001) (Table 4). A lower postoperative Harris hip score was correlated with COVID-19 infection (β: −8.20, 95% CI: −16.75 to −0.35, *p* = 0.048), a higher Charlson comorbidity index score (β: −1.78, 95% CI: −3.62 to −0.07, *p* = 0.049), and a lower preoperative Barthel index (β: 0.36, 95% CI: 0.15–0.58, *p* = 0.001) (Table 4). We investigated the factors which were associated with postoperative mortality. After adjustment for various factors, age (adjusted odd ratio (aOR): 1.25, 95% CI: 1.001–1.55, *p* = 0.049), gender (aOR: 342.45, 95% CI: 3.56–32910.44, *p* = 0.012), ASA grade (aOR: 28.96, 95% CI: 1.42–590.37, *p* = 0.029), and Charlson comorbidity index score (aOR: 0.22, 95% CI: 0.06–0.85, *p* = 0.028) were closely associated with postoperative mortality (Table 5).

## 4. Discussion

In our study, we noticed that time to operation did not significantly differ between the positive and negative groups (1.58 ± 0.51 vs. 1.38 ± 0.39 days, respectively). No significant differences were observed in the length of hospital stay or postoperative complications between the groups. Kim discovered that during the COVID-19 pandemic, hip fracture surgeries were postponed for 24 to 36 h, and the rate of postoperative complications did not increase [11]. Wang observed no change in the 30-day mortality rate, time to surgery, or length of hospital stay in a level-1 trauma center in the United States before, during, and after the COVID-19 pandemic [8]. However, a systematic review of 11 cohort studies and a total of 336 patients discovered that the in-hospital mortality rate of hip fracture patients was 29.8% (95% CI: 26.6−35.6%), the 30-day postoperative mortality rate was 35% (95% CI: 29.9−40.5%), and the average hospital stay was 11.29 days [12]. Another systematic review and meta-analysis discovered that patients with hip fractures who had concomitant COVID-19 infection had a 34% short-term mortality rate [13]. Mastan et al. found that COVID-19 status was associated with a 4-fold increase in mortality among patients with hip fractures [14]. In a study conducted by Raheman, patients with hip fractures who had COVID-19 had a 4-fold risk of mortality (risk ratio: 4.59, *p* < 0.0001), and the 30-day mortality rate was 38% (hazard ratio: 4.73, *p* < 0.0001). Raheman et al. discovered that male sex, diabetes, dementia, and extracapsular fractures were risk factors for mortality in patients with COVID-19 [15]. However, in our study, the mortality rate did not significantly differ between patients who had COVID-19 and those who did not have COVID-19. None of the seven patients who died had COVID-19.

The postoperative Barthel index score was significantly lower in the COVID-19-positive group (74.58 ± 17.64) than in the COVID-19-negative group. The postoperative Barthel index score and Harris hip score were associated with COVID-19 status (Table 4). Two factors may have contributed to this result. First, COVID-19 may have made the patients more vulnerable to health problems or caused their existing health problems to worsen. Second, the delayed initiation of rehabilitation may have had a significant impact on the surgical results because of the isolation policy for COVID-19 infection. Early mobilization is essential for the optimal postoperative management of patients with hip fractures, including activities such as getting in and out of bed, performing sit-to-stand exercises, rising from chairs with assistance, and walking with the aid of a walker [16,17,18,19]. Patients with hip fractures experience an average loss of muscle strength in their affected limbs of more than 50% in the first postoperative week [20,21,22,23].

The COVID-19 pandemic clearly impacted surgical treatment for patients with a hip pathology. Moldovan et al. quantified the effects of COVID-19 on elective arthroplasty interventions in Romania. He found that the COVID-19 pandemic had a severe impact on the volume of elective arthroplasty cases in Romania’s 120 hospitals [9], and this impact had significant financial ramifications for the hospitals. The author proposed the development of new clinical procedures and personalized home recovery programs for future outbreaks. Telemedicine through virtual consultations may also be integrated into emergency orthopedics in the future to maintain the care quality of patients during infectious disease pandemics [24,25]. The COVID-19 pandemic has placed unprecedented strain on healthcare systems worldwide, affecting various medical specialties, including orthopedics [26]. The pandemic apex has led to specific challenges in managing hip fractures among older adults in Taiwan in terms of both surgical quality and postoperative care. In terms of the impact of hip fractures on surgical quality, it can be divided into four parts: (1) Resource Allocation and Triage: The pandemic has forced many hospitals to reprioritize surgeries, with centers often postponing elective surgeries to focus on emergency cases [27]. In Taiwan, this has impacted the availability of resources like surgical suites, specialized orthopedic teams, and even equipment, which could potentially affect the surgical outcomes for hip fractures. (2) Surgical Delays: With hospitals at or near capacity due to COVID-19 patients, surgical delays have become common. A study in the *Journal of Bone and Joint Surgery* indicated that even a delay of just over 48 h could lead to a significant increase in 30-day mortality rates for hip fractures [28]. Delays may be exacerbated if the patient is COVID-19-positive, given the need for special protocols and isolation measures. (3) Surgical Technique and Team Experience: Due to the need for staff redeployment to care for COVID-19 patients, less experienced teams may sometimes handle surgeries, possibly affecting the surgical outcomes. Additionally, some studies suggest that less invasive surgical techniques might be preferred during pandemic conditions to reduce the operation time and hospital stay, although this could affect long-term outcomes [29]. (4) Perioperative Care: Special precautions have to be taken if the patient is COVID-19-positive. These include changes in anesthesia protocols and more intensive monitoring, which could affect the overall surgical experience and potentially lead to complications [30]. In terms of the impact on postoperative care quality for hip fractures, it can be divided into six parts: (1) Hospital Stay: During the pandemic’s apex, the focus was on reducing the length of hospital stays to free up beds. Quick discharge protocols may not always align with the optimal recovery paths for hip fracture patients, especially older adults, who often have comorbid conditions requiring complex care [9]. (2) Physical Rehabilitation: Given the social distancing norms and limitations on in-person interactions, physical rehabilitation schedules may be disrupted. A meta-analysis study in *Medicine* indicates that reduced postoperative mobility can lead to complications such as joint stiffness and an increased fall risk [31]. (3) Psychological Impact: Isolation due to COVID-19 protocols, coupled with the natural apprehension arising from being in a hospital during a pandemic, can lead to mental health issues like depression or anxiety [32]. These psychological factors can adversely affect postoperative recovery. (4) Follow-Up and Long-Term Care: Telehealth has often replaced in-person consultations for follow-up care, but not all aspects of postoperative care can be adequately managed remotely [24]. Moreover, older adults may face challenges in accessing or using digital platforms. (5) Economic Impact: In Taiwan, as elsewhere, the economic repercussions of the pandemic have led to funding cuts and resource allocation changes that could affect the quality of postoperative care. (6) Complications: According to an article in the *World Journal of Orthopedics*, the incidence of postoperative complications like pneumonia, urinary tract infections, or deep vein thrombosis may rise due to the strains and changes in standard care protocols during a pandemic [33]. As Taiwan navigates the challenges of COVID-19, innovative strategies like tele-rehabilitation, personalized home-based recovery plans, and the integration of artificial intelligence into the monitoring of postoperative care are becoming more relevant. Research and healthcare policies must adapt to ensure that the surgical and postoperative care quality of hip fracture patients is not compromised, irrespective of pandemic conditions. The apex of the COVID-19 pandemic has had a multifaceted impact on the quality of surgical and postoperative care for hip fractures in Taiwan. With shifting resources and strained healthcare systems, the implications are vast and warrant urgent attention to mitigate adverse outcomes.

The COVID-19 pandemic has significantly disrupted healthcare systems, and its impact extends beyond immediate medical care to longer-term functional outcomes in various patient groups. Patients recovering from surgeries often require extensive postoperative care, including physical therapy [31]. The pandemic has led to limitations in access to physical rehabilitation services due to social distancing measures and the diversion of healthcare resources. A study in *Clinical Rehabilitation* has indicated that this reduction in physical therapy access can have negative implications for long-term functional recovery, especially in the first 6 months post-surgery, which is a critical period for regaining function [34]. For patients with chronic conditions like COPD, diabetes, or heart disease, routine care and regular exercise are vital for maintaining functional ability. A study in *Experimental Gerontology* found that the interruption of regular healthcare visits and reduced physical activity due to lockdown measures can result in deteriorated functional quality over a 6-month period [35]. For instance, decreased exercise can exacerbate issues with glycemic control in diabetics, and reduced pulmonary rehabilitation can affect respiratory function in COPD patients. Regarding COVID-19 survivors, a number of studies have examined the “long COVID” phenomenon, where symptoms persist for months after the initial infection has resolved [36]. These symptoms can range from fatigue and muscle weakness to difficulties with concentration and memory, all affecting functional quality. A study in *The Lancet Global Health* found that even mild cases of COVID-19 can have lingering functional impacts up to 6 months after recovery, affecting the ability to return to work and carry out daily activities [37]. Elderly patients are at high risk of functional decline due to both age and an increased susceptibility to COVID-19. Social isolation measures, although necessary, have contributed to reduced physical activity, and studies in *The Journal of Frailty & Aging* have highlighted how this inactivity can lead to rapid functional decline in elderly populations [38]. The pandemic also had a widespread psychological impact, exacerbating conditions like depression and anxiety, which can have a direct effect on physical health and functional ability [39]. Psychological stress can influence pain perception, sleep quality, and overall well-being, factors that are critical in functional recovery from any illness or surgical intervention [32]. The above conditions may explain why the apex of the COVID-19 pandemic had a profound impact on the 6-month functional quality of patients with hip fractures in our study. Reduced access to healthcare services, interruptions in routine care, and the lingering effects of COVID-19 itself pose challenges that need urgent attention. Telemedicine and home-based care models are emerging as potential alternatives, but there is a dire need for more research in order to understand how to effectively mitigate these functional impairments.

This study has several limitations. First, this was a single-center study with a small sample (*n* = 85). The unexpected phenomenon whereby all the patients who died were in the COVID-19-negative group is due to the small sample size. Second, this study compared patients with and without COVID-19 during the pandemic and did not evaluate patients before the pandemic. Third, we did not include other parameters such as bone density, sarcopenia, and nutrition status that may have also impacted the patients’ postoperative function scores. In the future, we could design a matching study to reduce the bias from other interfering causes. The strength of our study is that it is based on real-world data obtained during patient admission and during follow-up in the outpatient clinic department or using telephone interviews. Few studies have investigated functional outcomes in patients with hip fractures who have COVID-19. Our study provides information on the association between functional outcomes and COVID-19 in patients with hip fractures.

## 5. Conclusions

At our hospital, orthopedic surgeries were not unduly delayed during the COVID-19 pandemic. No significant differences in the length of hospital stay, postoperative complications, or mortality were observed between patients with and without COVID-19. The key finding of this study is that among patients with hip fractures, those with COVID-19 may have worse functional outcomes in terms of the Barthel index and Harris hip score than those without COVID-19.

## Figures and Tables

**Figure 1 medicina-59-01640-f001:**
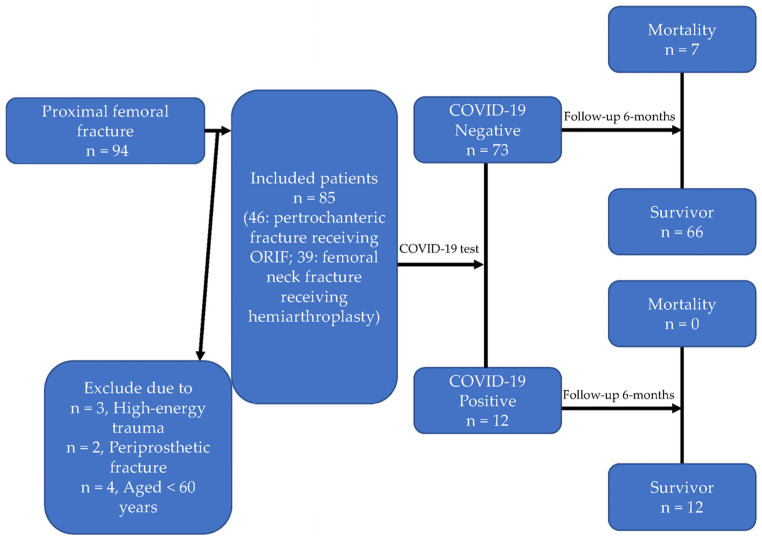
Flowchart of this study.

**Table 1 medicina-59-01640-t001:** Preoperative demographics of patients with proximal femoral fractures receiving ORIF (*n* = 85).

Variable	COVID-19 Infection		Total	*p*
	Negative	Positive		
N	73	12	85	
Age	79.18 ± 9.43	80.92 ± 10.44	79.42 ± 9.53	0.561
Body mass index (kg/m^2^)	22.42 ± 3.76	23.54 ± 3.48	22.58 ± 3.72	0.339
Gender	-	-	-	0.051
Male	28 (38.4%)	1 (8.3%)	29 (34.1%)	
Female	45 (61.6%)	11 (91.7%)	56 (65.9%)	
ASA physical status classification	-	-	-	0.755
1	7 (9.6%)	0 (0.0%)	7 (8.2%)	
2	25 (34.2%)	4 (33.3%)	29 (34.1%)	
3	31 (42.5%)	7 (58.3%)	38 (44.7%)	
4	10 (13.7%)	1 (8.3%)	11 (12.9%)	
Fracture type	-	-	-	0.119
Pertrochanteric	31 (42.5%)	8 (66.7%)	39 (45.9%)	
Femoral neck fracture	42 (57.5%)	4 (33.3%)	46 (54.1%)	
Preoperative blood test				
Hemoglobin (g/dL)	11.34 ± 2.12	10.86 ± 2.50	11.27 ± 2.17	0.478
Platelet (×10^3^/uL)	205.21 ± 64.4	218.83 ± 88.46	207.13 ± 67.85	0.522
PT (s)	11.10 ± 2.22	10.96 ± 0.63	11.08 ± 2.07	0.822
aPTT (s)	26.65 ± 4.26	28.97 ± 3.99	26.98 ± 4.27	0.082
INR	1.07 ± 0.22	1.07 ± 0.06	1.07 ± 0.21	0.949
ALT (U/L)	22.01 ± 18.43	28.83 ± 36.49	22.98 ± 21.71	0.316
BUN (mg/dL)	27.86 ± 21.61	16.67 ± 6.97	26.28 ± 20.55	0.080
Creatinine (mg/dL)	1.32 ± 1.39	0.75 ± 0.19	1.24 ± 1.30	0.164
Na (mmol/L)	137.16 ± 4.37	135.92 ± 6.73	136.71 ± 4.86	0.055
K (mmol/L)	5.76 ± 15.70	3.92 ± 0.63	5.50 ± 14.55	0.687
Charlson comorbidity index	5.26 ± 2.04	4.92 ± 1.93	5.21 ± 2.02	0.588
Comorbidity number	1.81 ± 1.25	2.25 ± 0.97	1.87 ± 1.22	0.248
Preoperative Barthel index	93.01 ± 14.71	93.75 ± 11.31	93.12 ± 14.23	0.869

Data are presented as *n* or mean ± standard deviation. aPTT: activated partial thromboplastin time; ASA: American Society of Anesthesiologists; PT: prothrombin time; INR: international normalized ratio.

**Table 2 medicina-59-01640-t002:** Perioperative demographics of patients with proximal femoral fractures receiving ORIF (*n* = 85).

Variable	COVID-19 Infection		Total	*p*
	Negative	Positive		
Surgical method	-	-	-	0.119
Hemiarthroplasty	42 (57.5%)	4 (33.3%)	46 (54.1%)	
ORIF	31 (42.5%)	8 (66.7%)	39 (45.9%)	
Anesthesia method	-	-	-	1.000
General	45 (61.6%)	8 (66.7%)	53 (62.4%)	
Neuroaxial	28 (38.4%)	4 (33.3%)	32 (37.6%)	
Perioperative nerve block	57 (78.1%)	11 (91.7%)	68 (80.0%)	0.445
Preoperative NPO time (h)	12.18 ± 3.34	13.00 ± 3.81	12.29 ± 3.40	0.441
Time to operation (day)	1.38 ± 0.49	1.58 ± 0.51	1.41 ± 0.50	0.197
Operation time (min)	67.34 ± 20.92	73.83 ± 22.76	68.26 ± 21.17	0.328
Anesthesia time (min)	137.73 ± 32.30	158.25 ± 40.24	140.62 ± 34.03	0.046
Blood loss (mL)	190.41 ± 135.61	245.83 ± 178.96	198.24 ± 142.60	0.214
Postoperative hemoglobin level (g/dL)	10.07 ± 1.81	9.85 ± 2.03	10.04 ± 1.83	0.696

Data are presented as *n* or mean ± standard deviation ORIF = open reduction and internal fixation.

**Table 3 medicina-59-01640-t003:** Postoperative demographics of patients with proximal femoral fractures receiving ORIF (*n* = 85).

Variable	COVID-19 Infection		Total	*p*
	Negative	Positive		
Length of stay (day)	11.00 ± 9.85	11.08 ± 5.16	11.01 ± 9.31	0.977
Postoperative complication numbers	1.00 ± 0.91	0.92 ± 1.24	0.99 ± 0.96	0.782
Postoperative ECOG performance status	1.79 ± 0.83	2.00 ± 0.74	1.82 ± 0.82	0.412
Postoperative Barthel index	88.03 ± 16.82	74.58 ± 17.64	85.96 ± 17.53	0.014 *
Postoperative modified Harrison hip score	73.42 ± 14.01	66.00 ± 11.79	72.28 ± 13.89	0.089
Mortality	7 (9.6%)	0 (0.0%)	7 (8.2%)	0.586

Data are presented as *n* or mean ± standard deviation. * *p* < 0.05 was considered statistically significant after the test.

**Table 4 medicina-59-01640-t004:** Factors associated with postoperative complications, ECOG performance status, barthel index and modified Harrison hip score at postoperative 6 months (*n* = 85).

Variable	Incidence of Complications		ECOG Status		Bathal Index		HHS	
	β (95% CI)	*p*	β (95% CI)	*p*	β (95% CI)	*p*	β (95% CI)	*p*
Age	0.004 (−0.021, 0.029)	0.762	0.004 (−0.016, 0.023)	0.699	0.02 (−0.36, 0.40)	0.906	−0.09 (−0.46, 0.28)	0.637
Gender (Male vs. Female)	−0.09 (−0.53, 0.35)	0.689	0.06 (−0.30, 0.42)	0.732	−0.16 (−7.19, 6.86)	0.963	−3.71 (−10.62, 3.20)	0.287
COVID-19 infection (Positive vs. Negative)	−0.19 (−0.78, 0.40)	0.523	0.37 (−0.07, 0.82)	0.098	−15.59 (−24.29, −6.89)	0.001 *	−8.20 (−16.75, −0.35)	0.048 *
ASA physical status classification	0.13 (−0.18, 0.43)	0.408	−0.07 (−0.31, 0.17)	0.567	4.62 (−0.10, 9.34)	0.055	1.77 (−2.87, 6.41)	0.449
Fracture type (Pertrochanteric vs. Femoral neck)	0.08 (−0.35, 0.51)	0.725	0.04 (−0.30, 0.37)	0.829	−3.22 (−9.73, 3.28)	0.326	−1.98 (−8.38, 4.41)	0.538
Anesthesia (General vs. Neuraxial)	0.50 (0.06, 0.94)	0.026 *	0.14 (−0.20, 0.48)	0.416	−1.74 (−8.47, 5.00)	0.608	−3.95 (−10.57, 2.67)	0.238
Peripheral Nerve block (Yes vs. No)	0.11 (−0.40, 0.63)	0.660	0.03 (−0.38, 0.44)	0.882	−2.25 (−10.23, 5.72)	0.575	−3.06 (−10.89, 4.78)	0.439
Charlson Comorbidity Index	0.08 (−0.05, 0.20)	0.238	0.08 (−0.02, 0.18)	0.099	−3.30 (−5.18, −1.42)	0.001 *	−1.78 (−3.62, −0.07)	0.049 *
Preoperative Barthel Index	0.01 (−0.01, 0.02)	0.194	−0.03 (−0.04, −0.02)	<0.001*	0.65 (0.43, 0.87)	<0.001 *	0.36 (0.15, 0.58)	0.001 *
Time to operation	−0.02 (−0.46, 0.42)	0.943	−0.31 (−0.65, 0.03)	0.073	1.18 (−5.51, 7.87)	0.726	2.29 (−4.29, 8.86)	0.489
Operation time	−0.004 (−0.014, 0.006)	0.473	−0.004 (−0.012, 0.003)	0.266	−0.04 (−0.19, 0.12)	0.651	−0.03 (−0.18, 0.12)	0.674
Blood loss	0.001 (−0.001, 0.002)	0.411	−0.001 (−0.002, 0.001)	0.439	0.01 (−0.01, 0.04)	0.325	−0.004 (−0.029, 0.021)	0.726
Length of Stay	0.04 (0.01, 0.06)	0.004 *	0.01 (−0.01, 0.03)	0.195	0.01 (−0.37, 0.39)	0.958	0.00 (−0.37, 0.37)	0.998

Data are presented as β (95% CI). * *p* value < 0.05 was considered statistically significant after test. ASA = American Society of Anaesthesiologists, ECOG = Eastern Cooperative Oncology Group, HHS = modified Harrison hip score.

**Table 5 medicina-59-01640-t005:** Factors associated with mortality at 6 postoperative months (*n* = 85).

Variable	Crude		Adjusted	
	OR (95% CI)	*p*	OR (95% CI)	*p*
Age	1.03 (0.94, 1.12)	0.533	1.25 (1.001, 1.55)	0.049 *
Gender (Male vs. Female)	14.35 (1.63, 125.94)	0.016 *	342.45 (3.56, 32,910.44)	0.012 *
COVID-19 infection (Positive vs. Negative)	0.00 (NA)	0.999		
ASA physical status classification	2.60 (0.87, 7.74)	0.086	28.96 (1.42, 590.37)	0.029 *
Fracture type (Pertrochanteric vs. Femoral neck)	0.88 (0.18, 4.17)	0.867	3.48 (0.13, 95.13)	0.460
Anesthesia (General vs. Neuraxial)	2.46 × 10^8^ (NA)	0.998		
Peripheral nerve block (Yes vs. No)	0.60 (0.11, 3.37)	0.558	8.88 (0.16, 505.33)	0.289
Charlson comorbidity index	0.98 (0.66, 1.45)	0.924	0.22 (0.06, 0.85)	0.028 *
Preoperative Barthel index	0.99 (0.94, 1.04)	0.742	1.03 (0.91, 1.16)	0.663
Time to operation	2.02 (0.42, 9.66)	0.378	0.23 (0.01, 6.44)	0.390
Operation time	0.98 (0.94, 1.02)	0.355	0.97 (0.90, 1.05)	0.486
Blood loss	1.001 (0.996, 1.006)	0.672	1.01 (0.998, 1.02)	0.101
Length of stay	1.04 (0.98, 1.10)	0.170	1.06 (0.93, 1.20)	0.390

Data are presented as odds ratio (95% CI). * *p* value < 0.05 was considered statistically significant after the test. ASA: American Society of Anesthesiologists. NA: not applicable.

## Data Availability

The data are contained within the article.

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
