# Peer review of "COVID-19 Infection Was Associated with the Functional Outcomes of Hip Fracture among Older Adults during the COVID-19 Pandemic Apex"

_medicina, 2023, doi:10.3390/medicina59091640_

Round 1

Reviewer 1 Report

1-"Incidence of postoperative complications was significantly correlated with general anesthesia and length of stay in hospital. Poor postoperative functional scores were associated with lower preoperative Barthel Index scores. Male, old age, high American Society of Anesthesiologists grade, and high Charlson Comorbidity Index (CCI) score were associated with mortality. " Please provide relevant numerical values i.e. correlaton coefficients. 

2-" but their postoperative Barthel index scores were significant influenced by COVID-19 (+)" Could you exclude other interferers? 

3-Please use MeSH related keywords. 

4-The introduction should introduce the novelty of the study clearly. Please provide clear aim and hypotesis sentences. 

5-The methodology stands weak. 

6-How was the sample size arrived at? 

Minor editing would be required. 

Author Response

Response to Review 1

Comments and Suggestions for Authors

1-"Incidence of postoperative complications was significantly correlated with general anesthesia and length of stay in hospital. Poor postoperative functional scores were associated with lower preoperative Barthel Index scores. Male, old age, high American Society of Anesthesiologists grade, and high Charlson Comorbidity Index score were associated with mortality. " Please provide relevant numerical values i.e., correlation coefficients.

Ans: Thank you for your reminding. I have added the p-values as below:” Incidence of postoperative complications was significantly correlated with general anesthesia (p = 0.026) and length of stay (p = 0.004) in hospital. Poor postoperative functional scores were associated with lower preoperative Barthel Index scores (p < 0.001). Male (p = 0.049), old age (p = 0.012), high American Society of Anesthesiologists grade (p = 0.029), and high Charlson Comorbidity Index score (p = 0.028) were associated with mortality.

2-" but their postoperative Barthel index scores were significant influenced by COVID-19 (+)" Could you exclude other interferers?

Ans: Thank you for your reminding. We applied multiple linear regression analysis was applied to evaluate the associations of covariates and postoperative comorbidities with changes in postoperative Barthel index scores and found a significant association between COVID-19 infection and postoperative Barthel index scores. We have added below sentence into our limitation: “Third, we did not include other parameters such as bone density, sarcopenia, and nutrition status that may also have impact on the patients' postoperative function scores.” Further may design a matching study to reduce the bias of other cause that interferes.

3-Please use MeSH related keywords.

Ans: Thank you for your suggestion. We have modified our MeSH related keywords as below: COVID-19 ; hip fracture ; postoperative complications; activities of daily living; mortality.

4-The introduction should introduce the novelty of the study clearly. Please provide clear aim and hypothesis sentences.

Ans: Thank you for your suggestion. We have modified our last paragraph in introduction section as below: “We hypothesize that the COVID-19 pandemic has launched policies that will affect hip fracture postoperative care. It includes nursing care, family care, a decrease of early bedside rehabilitation programs which may further increase mortality and poor functional outcome. Our study was aimed at comparing mortality and functional outcomes between patients with and without COVID-19 after surgery for hip fracture in our medical center during COVID-19 pandemic apex in Taiwan.”

5-The methodology stands weak.

Ans: Thank you for your reminding. We have revised our Method section as below.

 Study Design and Population: We conducted a prospective cohort study at Hualien Tzu Chi Hospital, enrolling patients aged 60 and above who were diagnosed with hip fractures between March 9, 2022, and September 9, 2022. Patients with high-energy trauma, periprosthetic or peri-implant fractures, those who did not undergo surgery, or underwent revision surgery for a prior hip fracture were excluded from the study.

 Data Collection: Demographic and clinical data were meticulously extracted from electronic medical records. Preoperative functional scores were assessed upon patient admission through validated tools. Follow-up postoperative functional scores were obtained during scheduled outpatient consultations or structured telephone interviews conducted by trained medical staff. The parameters were listed as below: (1) Preoperative data: age, gender, body mass index (BMI), fracture type, American Society of Anesthesiologists (ASA) grade, blood chemistry, Charlson Comorbidity index (CCI), Barthel Index (BI) score, Eastern Cooperative Oncology Group (ECOG) score, and COVID-19 status, as confirmed by polymerase chain reaction tests; (2) Operative data: surgical approach, anesthesia type, use of nerve block, preoperative NPO time, time from admission to operation, duration of the operation, anesthesia time, estimated blood loss, and immediate postoperative hemoglobin levels; (3) Postoperative data: length of hospital stay (LOS), complication incidence (e.g., pneumonia, sepsis, acute urinary retention, ileus, electrolyte imbalance, urinary tract infection, anemia, moderate to severe hip pain), mortality, and postoperative functional scores (ECOG, BI, and modified Harris hip score [HHS]).

 Statistical Analysis: All statistical analyses were performed using SPSS version 23.0 (IBM, Armonk, NY, USA). Descriptive Statistics included continuous variables presented as means ± standard deviations (SD), and categorical variables shown as numbers and percentages. Univariate Analysis included Student's t-test employed for comparing continuous variables between groups and chi-square tests used for categorical variables. Multivariable Analysis included multiple linear regression analysis conducted to assess the impact of covariates and postoperative comorbidities on changes in ECOG, BI, and HHS scores, with adjustments made for potential confounders. Variable selection was based on a stepwise backward elimination process. Outcome analysis included logistic regression models used to identify factors independently associated with mortality, adjusted for confounding variables. Odds ratios and 95% confidence intervals were reported for all logistic regression analyses. Statistical significance was defined as a p-value of < 0.05, and all tests were two-tailed.

6-How was the sample size arrived at?

Ans: Thank you for your asking. Our study is a single medical center in the East of Taiwan relative has a low population, our collected sample within half year is only n = 85 only. If need to use incidence and relative risk to calculates the sample size required for a cohort in this topic is far away not enough. But we hope that after publication it can lead to further academic investigation the functional outcome associated with COVID-19 infection. COVID-19 becomes a usual medical problem in the global health system, we need to clearly identify the effect of COVID-19 on older adults with hip fractures, for improved functional outcomes.

Reviewer 2 Report

What is the main question addressed by the research?

Does COVID-19 infection have some impact on mortality and functional outcomes of patients with hip fractures?

2.      Do you consider the topic original or relevant in the field? Does it address a specific gap in the field?

Yes.  Whether or not the inferior quality of care during COVID-19 pandemic has resulted in increased mortality, adverse outcomes, and poor functional improvement among COVID-19 infected patients compared to non-infected patients.

3.      What does it add to the subject area compared with other published material?

It provides comparative evidence between patients with established COVID-19 infections and those without infection.

4.      What specific improvements should the author consider regarding methodology? What further controls should be considered?

None

5.      Are the conclusion consistent with the evidence and argument presented and do they address the main question posed?

Yes.

6.      Are the reference appropriate?

Yes. they seem appropriate

7.      Please include any comment on the table and figures?

They are clear and understandable

Author Response

Response to Reviewer 2

Comments and Suggestions for Authors

  1. What is the main question addressed by the research?

Does COVID-19 infection have some impact on mortality and functional outcomes of patients with hip fractures?

  1. Do you consider the topic original or relevant in the field? Does it address a specific gap in the field?

Yes.  Whether or not the inferior quality of care during COVID-19 pandemic has resulted in increased mortality, adverse outcomes, and poor functional improvement among COVID-19 infected patients compared to non-infected patients.

  1. What does it add to the subject area compared with other published material?

It provides comparative evidence between patients with established COVID-19 infections and those without infection.

  1. What specific improvements should the author consider regarding methodology? What further controls should be considered?

None

  1. Is the conclusion consistent with the evidence and argument presented and do they address the main question posed?

Yes.

  1. Is the reference appropriate?

Yes. they seem appropriate

  1. Please include any comment on the table and figures.

They are clear and understandable.

Ans: Thank you for your encouragement very much. We will male greater effort working on this topic for improvement of patients’ care quality.

Reviewer 3 Report

The research aim is to analyze the influence of COVID-19 infection on hip fractures outcomes after surgical treatment. The  article is interesting and the subject not yet fully investigated.

The abstract is structured appropriately.  

The introduction transposes the research into the topic and formulates the objective of the study at the end. However,  more precise information about the impact of COVID-19 pandemic on the volumes of hip arthroplasties should be added in relation to other scientific papers, for e.g. Moldovan, F.; Gligor, A.; Moldovan, L.; Bataga, T. An Investigation for Future Practice of Elective Hip and Knee Arthroplasties during COVID-19 in Romania. Medicina 2023, 59, 314. doi: 10.3390/medicina59020314

In the methodology section, the stages of the research are presented but a flow diagram for selection of the study population could be provided for a better overview. Also, the surgical procedures performed should be described as it not clear how the pertrochanteric and femoral neck fractures were treated.

The results are clearly described. In Table 1, the sample size should be included in the header of the table (as n value).

The discussions interpret the research results and relate them to other results from the scientific literature. The limitations of the study should be presented in a different paragraph.

The conclusions are concise and clear.

The bibliography is adequate and properly edited but can be extended as suggested above.

Author Response

Response to Review 3

Comments and Suggestions for Authors

The research aim is to analyze the influence of COVID-19 infection on hip fractures outcomes after surgical treatment. The article is interesting and the subject not yet fully investigated. The abstract is structured appropriately. 

Ans: Thank you for your encouragement. We have supplemented the content by your suggestions to make our article more thorough.

The introduction transposes the research into the topic and formulates the objective of the study at the end. However, more precise information about the impact of COVID-19 pandemic on the volumes of hip arthroplasties should be added in relation to other scientific papers, for e.g. Moldovan, F.; Gligor, A.; Moldovan, L.; Bataga, T. An Investigation for Future Practice of Elective Hip and Knee Arthroplasties during COVID-19 in Romania. Medicina 2023, 59, 314. doi: 10.3390/medicina59020314

Ans: Thank you for your suggestions. We have added the content as below in Introduction section:” Moldovan et al conducted a multicenter study in Romania in 2023 found that the COVID-19 pandemic has had a severe impact on the volume of elective arthroplasty in Romania's 120 hospitals and was a dramatic decrease in the volume of primary interventions in hip and knee patients by up to 69.14%, with a corresponding decline in the quality of patient care [9] .” We also have added the related discussion in Discussion section about correlation of hip surgeries, postoperative care and COVID-19 infections.

In the methodology section, the stages of the research are presented but a flow diagram for selection of the study population could be provided for a better overview. Also, the surgical procedures performed should be described as it does not clear how the pertrochanteric and femoral neck fractures were treated.

Ans: Thank you for your suggestions. We have added the flowchart below as Figure 1. We have added the surgical procedures as below into our Material section:” The patients with pertrochanteric fracture received open reduction and internal fixation with cephalomedullary nail, while the patients with displaced femoral neck fracture received hemiarthroplasty.”

The results are clearly described. In Table 1, the sample size should be included in the header of the table (as n value).

Ans: Thank you for your reminding. I have added this value.

The discussions interpret the research results and relate them to other results from the scientific literature. The limitations of the study should be presented in a different paragraph.

Ans: Thank you for your reminding. We have made this change.

The conclusions are concise and clear.

The bibliography is adequate and properly edited but can be extended as suggested above.

Ans: Thank you for your suggestions. We have modified our manuscript as your suggestions.
